# Levels of Physical Activity in Children with Extremity Fractures a Dutch Observational Cross-Sectional Study

**DOI:** 10.3390/children9030325

**Published:** 2022-03-01

**Authors:** Amber Carlijn Traa, Ozcan Sir, Sanne W. T. Frazer, Brigitte van de Kerkhof-van Bon, Birgitte Blatter, Edward C. T. H. Tan

**Affiliations:** 1Department of Emergency Medicine, Radboud University Medical Center, Geert Grooteplein Zuid 10, 6525 GA Nijmegen, The Netherlands; ozcan.sir@radboudumc.nl (O.S.); edward.tan@radboudumc.nl (E.C.T.H.T.); 2Consumer Safety Institute (VeiligheidNL), Overschiestraat 65, 1062 XD Amsterdam, The Netherlands; s.frazer@veiligheid.nl (S.W.T.F.); b.blatter@veiligheid.nl (B.B.); 3Department of Emergency Medicine, Canisius Wilhelmina Hospital, Weg door Jonkerbos 100, 6532 SZ Nijmegen, The Netherlands; b.vandekerkhofvanbon@cwz.nl; 4Department of Traumasurgery, Radboud University Medical Center, Geert Grooteplein Zuid 10, 6525 GA Nijmegen, The Netherlands

**Keywords:** injury prevention, fracture risk, global recommendations on physical activity for health, SQUASH questionnaire, multi-center

## Abstract

Background: Fractures are common in children and a frequent cause of emergency department (ED) visits. Fractures can cause long-term complications, such as growth problems. Research on fractures can reveal useful areas of focus for injury prevention. Objective: To assess the role of physical activity in the occurrence of fractures, this study investigates physical activity among children with extremity fractures based on the Global Recommendations on Physical Activity for Health. Methods: A multi-center, cross-sectional study was performed at two EDs in Nijmegen, the Netherlands. Patients between 4 and 18 years of age visiting these EDs with a fracture were asked to complete a validated questionnaire. Results: Of the 188 respondents, 51% were found to adhere to the recommendations. Among participants between 13 and 18 years of age, 43% were adequately physically active, compared to participants between 4 and 12 years of age among whom 56% were adequately physically active (*p* = 0.080). Additionally, more males were found to meet the recommendations (60% versus 40%). The most common traumas were sports-related (57%). Sports-related traumas were cited more often among youth between 13 and 18 years of age, compared to those between 4 and 12 (*p* < 0.001). Conclusions: A relatively high prevalence of adherence to the Global Recommendations on Physical Activity for Health was observed among children with fractures. Most respondents obtained their fractures during participation in sports. This study emphasizes the need for more injury prevention, especially among youth between 13 and 18 years of age and children participating in sports.

## 1. Introduction

The annual rate of fractures among children has been reported as ranging between 12 to 36 for every thousand children [1,2,3]. Fractures can result in a decrease in physical activity (PA) and long-term complications, such as growth problems. Studying fractures in children could support in developing preventive measures reducing fracture rates and avoid the possible negative consequences for children’s health [4,5,6]. 

Previous studies have concluded that PA during childhood is associated with higher bone mineral density and long-term muscular benefits [7,8]. Additionally, there have been suggestions that increased PA decreases the incidence of fractures in children [9,10]. However, children’s PA can also lead to injuries. Since children often take risks while playing and are thus at risk of being injured, PA could also lead to fractures [11,12]. 

In an effort to enhance global health systems and prevent curable diseases, the World Health Organization (WHO) developed the Global Recommendations on Physical Activity for Health in 2010. Revised in 2020, these universal recommendations for physical activity specifically target children between 5 and 17 years of age. The WHO recommends that every child exercise for at least one hour a day at a moderate-to-intensive level of intensity and also perform muscle- and bone-strengthening exercises at least three times a week. Since their publication, these recommendations have been widely adopted. For example, the United States (US) and the European Union (EU) have adopted these recommendations created by the WHO [13,14].

The WHO found that, globally, adolescents between 11 and 17 years of age attending school are insufficiently physically active in 81% of documented cases, meaning that 19% of adolescents were adhering to the WHO’s recommendations [15]. In the EU, sufficient levels of PA according to the WHO’s recommendations varied from 17% to 43% of cases [14]. In the Netherlands, the National Institute for Public Health and the Environment found that 42% of Dutch children between 5 and 17 years of age were sufficiently physically activity [16]. In general, the percentage of sufficient PA in children according to the WHO’s recommendations is less than 50%. 

Multiple studies have found fracture rates to be significantly higher among males compared to females [17,18]. Moreover, different trauma mechanisms during childhood have been observed in different age groups (1). A deeper investigation into fracture rates in children in relation to trauma mechanisms and PA could generate insights into the factors contributing to fractures. These findings would support the development of preventive measures by revealing the risk factors related to fractures during childhood. 

This study explores the Global Recommendations on Physical Activity for Health created by the WHO by focusing on a population of children with extremity fractures from the Netherlands. This study aims to develop insights into levels of PA among children with extremity fractures based on the WHO’s recommendations. This study also investigates the trauma mechanisms causing fractures in children. 

## 2. Methods and Materials

### 2.1. Study Design and Setting

This study was designed with a multi-center, cross-sectional approach. The study was conducted at the emergency departments (EDs) of the Radboud University Medical Center (Radboudumc) and Canisius Wilhelmina Hospital (CWZ). Both centers are located in Nijmegen, the Netherlands. Radboudumc is a level 1 trauma center, and CWZ is a level 2 trauma center. Combined, the hospitals receive approximately 59,000 patient visits yearly. Both participating hospitals are located in Nijmegen, the Netherlands. Nijmegen is a mid-urbanized city in a rural environment and as such, is considered representative of the Netherlands. The Netherlands is a developed country and member of the EU. Prior to the data collection, the study was assessed and accepted by the Institutional Review Board of both Radboudumc and CWZ. Informed consent was obtained from all parents or the patient when older than 16.

### 2.2. Study Population

The sample for this study included all patients between 4 and 18 years of age at the time of their visit to the EDs of the Radboudumc or CWZ with a fracture acquired between November 2017 and November 2018. Patients were only included if their fracture was radiologically proven. All participating children and their parents or guardians received paper questionnaires in December 2018. In January 2019, in cases of four to six weeks non-response, the researcher called patients with a request to participate in the study in order to enlarge the response rate. Patients were excluded if they had specific diseases, were taking medications that affect bone metabolism, were involved in a high-impact trauma, or were diagnosed with more than two fractures or a fracture of the skull, face, thorax or pelvis. All exclusion criteria are specifically outlined in the questionnaire (Appendix A). An a priori power size calculation indicated that, in total, 175 patients had to be included for there to be a sufficient sample size with a power of 0.8 at an alpha level of 0.05. This calculation involved the prevalence of Dutch children meeting sufficient PA levels presented in previous studies, namely 42% [16]. 

### 2.3. Data Collection

Patients received study information, an informed consent letter, and a coded questionnaire by mail. The questionnaire and informed consent letter were to be sent back to the Radboudumc for analysis. The questionnaire was developed based on previous validated questionnaires. For an evaluation of patients’ level of PA, the Short Questionnaire to Assess Health Enhancing Activity (SQUASH) was used [19]. This validated questionnaire was also used for national research on the WHO’s recommendations among adults and children [20]. The SQUASH questionnaire has been validated in adults and adolescents [21]. Along with the SQUASH questionnaire, patients were asked to report on the event that caused their injury and the amount of time spent behind a screen. The intensity level of the activity performed was calculated based on Metabolic Equivalents of Task (METs) (Appendix A). Calculating METs is an objective way to measure the intensity of PA performed by the respondents.

### 2.4. Data Analysis

All data processing and analyses were performed with Microsoft Office Excel 2016 (Microsoft Corp., Redmond, WA, USA), Castor EDC (Ciwit B.V., Amsterdam, the Netherlands) and SPSS Statistics for Windows, version 25 (IBM Corp., Armonk, NY, USA). The data were described in means and percentages with standard deviations (SDs). The main variable was adherence to the recommendations created by the WHO (yes/no). This study used the same method of data analysis for the WHO’s recommendations as was used for the national study on this topic performed by the National Institute for Public Health and the Environment [16]. The two sub-measurements of the norm were separately analyzed and then combined for the researcher to determine whether the recommendations were being adhered to. The sub-measurements calculated whether respondents: (1) exercise for at least one hour a day at a moderate-to-intensive level of intensity (2) perform muscle- and bone-strengthening exercises at least three times a week. Pearson χ2 tests, ANOVAs (*F*), one-sample, and independent *t*-tests were conducted for a comparison between age groups and gender. *p*-values lower than 0.050 were considered statistically significant. Missing data are either mentioned separately in the results section or represented in the number of individuals (*n*) for each result. Analyzation of age groups 4 to 12 years and 13 to 18 years of age was performed since it was expected to find differences in PA. Therefore, calculations on solely the total study population could be misleading. Secondly, in the Netherlands, the National Institute for Public Health and the Environment also performed statistics on age subgroups making interpretations of the study results in relation to these data more insightful.

## 3. Results

### 3.1. Study Population and Baseline Characteristics

In total, 1107 potential respondents received questionnaires. Of these potential respondents, 216 had visited the ED of the Radboudumc and 891 had visited the ED of the CWZ. The total response rate was 20% (*n* = 216) (Radboudumc 24%, CWZ 18%). Of the respondents, 13% (*n* = 28) were excluded according to the exclusion criteria. In total, 188 patients were included in this study’s analysis, as outlined in Figure 1. 

The mean age was 11.2 (SD = 3.7, Range 4–18), and 57.4% (*n* = 108) of the respondents are male. The baseline characteristics of patients included in this study are similar compared to non-respondents (*p* > 0.600). More details are included in Table 1.

### 3.2. PA and the WHO’s Global Recommendations on Physical Activity for Health

The mean time spent on moderate-to-intensive activities was 2.5 h daily (Table 2). With the exclusion of physical education, 84% of the children (*n* = 157) were actively participating in at least one sport at the time of the study. Moreover, children not meeting the physical activity recommendations for health spent significantly less time on sports compared to children meeting these recommendations (Table 3). Youth between 13 and 18 years of age were found to spend more time playing sports (*p* < 0.001). In contrast, young children were found to play outside more frequently than respondents between 13 and 18 years of age (*p* < 0.005). 

Of all the children included in this study, 50.5% (*n* = 95) were found to exercise for at least 1 h at a moderate-to-intensive level of intensity daily. This study also found that 99.4% (*n* = 187) of all children perform muscle- and bone-strengthening exercises at least three times a week. In total, 50.5% (*n* = 95) of the respondents were meeting the criteria outlined in the WHO’s Recommendations on Physical Activity for Health. Children between 4 and 12 years of age were more likely to be adhering to the WHO’s recommendations compared to those between 13 and 18 years of age; however, this was not statistically significant. More details are included in Table 2 and Table 3. Table 3 presents a comparison of children meeting the recommendations for health versus children not meeting these recommendations. There are no significant differences between both groups, with only children meeting the recommendations on physical health spending more time on sports and having a decreased screen time compared to children not adhering to the recommendations. Physically active children spent 191 min per day (SD = 131) on screentime whereas physically inactive children spent 220 (SD = 171) minutes per day behind a screen.

### 3.3. Trauma Mechanism and Type of Fracture

Most children had acquired one fracture, namely 83% (*n* = 156). The rest of the study population acquired two fractures. Additionally, an upper extremity fracture was most often diagnosed, found in 82% (*n* = 154). Of the fractures, 56% (*n* = 105) occurred during sports. Sports-related traumas occurred significantly more often in youth between 13 and 18 years of age than children between 4 and 12 (*p* < 0.001). In contrast, a fall between 0.5 and 3 m occurred more frequently in children between 4 and 12 years of age compared to youth between 13 and 18 (*p* < 0.001) (Table 3).

Twenty-three percent (*n* = 43) of the children reported that the trauma happened while playing soccer, and 17% (*n* = 32) of the fractures were caused by a trauma involving climbing a frame, play set, or rack. Figure 2 provides more details on the trauma mechanisms found among children.

## 4. Discussion

This study found a relatively high prevalence of adherence to the WHO’s Global Recommendations on Physical Activity for Health among children with fractures. The results demonstrate that 51% of the children with fractures involved in low to medium impact trauma adhere to the WHO’s recommendations. Older children adhered to the recommendations less than younger children. The most common traumas in children with fractures were sports-related. 

A comparison of this study’s results and the general national study conducted on the WHO’s recommendations reveals that there is an approximate 7% difference in adherence to the recommendations between Dutch children with fractures and Dutch children in general (50% versus 42%) [16]. The difference in adherence to the WHO’s recommendations is even larger when this study is compared to the European and global literature which reports adherence to the WHO’s recommendations in 19–43% of children [13,14,15,22]. However, it must be noted that differences in study methodologies between this study and the international literature with regard to response bias could have influenced the results. Overall, comparing and generalizing the study results to global literature gives insight but should be interpreted with substantial caution. 

Although a relatively high prevalence of children adhering the WHO’s recommendations in this study was found, this study did not find sufficient evidence to suggests that physically active children are more likely to obtain a fracture. In existing literature, contradicting results have been described regarding fracture risk and PA [9,10,12,17,23]. The differences in results could be explained by the fact that there is a contradiction in the influence of PA on fracture risk. PA both benefits bone mineral density, thus reducing fracture risk, and involves situations that potentially increase injury rates [18]. This study’s results support the rationale that PA increases the exposure to risky situations, such as sports or playing [11,12]. Also, this study found boys to adhere more frequently to the WHO’s recommendations compared to girls. This is in accordance with the national literature [24]. This is potentially explained by children’s, especially boys’, tendency to participate in risky situations as a part of their upbringing and development [11]. This is underlined in this study through a high participation in sports at an older age. Overall, this study has found a relatively high prevalence of children adhering to the WHO’s recommendations, which supports the idea that PA increases exposure to the risk of injuries. However, differences in study results compared to national literature were too small to state that PA increases fracture risk. 

Although PA might lead to more exposure to risky situations, it is important to acknowledge the positive effects of PA on social, psychological, and physical health [25,26]. The necessity of preventive actions is underlined by this study’s finding that especially younger children with fractures are very physically active. As stated in the World report on child injury prevention, preventive measures including the redesign of playing equipment, legislative enforcements, and multifaceted community programs have been found effective in reducing injury risk [27,28,29]; combining these measures is even more effective. Examples of preventive measures are regulating playground equipment standards, supportive home visitation to identify fall hazards, and investment in safer routes to school [27]. Behavioral changes, such as safer playing behavior, can also prevent injuries. Evidence suggests that one of the methods for achieving behavioral changes is educating parents about their children’s risky playing behavior [30]. Preventive measures and behavioral changes should be taken into account in efforts to decrease fracture rates in children. These preventive measures should, as suggested by the study results, be targeted at adolescents and children participating actively in sports. 

In line with national and international literature, this study observed age differences in the percentages of adherence to the WHO’s recommendations [16]. Children between 4 and 12 years of age were adequately physically active in 57% of the cases, compared to 43% among children between 13 and 18. The trend of younger children more frequently adhering to the WHO’s recommendations on PA compared to older children is supported by previous studies on PA among children [14,16]. This trend can be explained by the sedentary behavior and long periods of time spent behind screens among adolescents, also found in this study [31]. 

Additionally, this study has found the most common traumas in low to medium impact trauma to be sports-related. This is confirmed by international literature [32,33]. Moreover, 84% of the children with fractures, in this study, were found to be active in sports. This is more than the 70.3% found in the general pediatric Dutch population [34]. Statistics also suggest that 80% of Dutch children are actively involved in sports clubs of schools [35]. The finding that children with fractures are more active in organized PA such as sports supports the idea that PA exposes children to the risk of injury and possibly increases fracture risk [11,12]. 

This study has also found that among younger children, a fall of 0.5 to 3 m occurred significantly more frequently, in contrast to older children, who were more frequently involved in sports traumas. This is supported by European and American literature on different fracture etiologies in different age groups [36]. Finally, 23% of the fractures described in the questionnaires were soccer related, which is, according to European literature, representative of national sport preferences [36]. In the Netherlands, soccer is a popular sport; reports have found that 22.6% of Dutch children are reported to play soccer [34]. However, it is possible that by solely including extremity fractures instead of all fractures of the body, this study has overestimated the impact of PA on fracture rates. The results on trauma mechanisms present patterns based on age and national sports preferences. As addressed before, preventive measures could help to decrease fracture incidence among children since the occurrence of trauma during sports has been found to be high [37,38]. 

A major limitation of this study is the low response rate of 20%. However, a sufficient sample size was achieved, and the respondent versus non-respondent analysis found similar baseline characteristics. In other survey studies, response rates vary widely, ranging from 20% to 43% [39,40]. Bias could have also influenced the results. Participation and response bias could have been present since physically active children and children in higher socio-economic classes are more likely to respond [18]. Additionally, the study’s design is prone to recall bias, as it is a retrospective study which requires the respondent to fill in a questionnaire on an event that might have occurred up to one year in the past. In addition, the SQUASH questionnaire was validated for adults and adolescents yet has not been validated for younger children. It was expected that for younger children the parents or guardian filled out the questionnaire. This was also addressed in information letters provided for participants. However, response bias could have occurred. The inclusion of potential participants over one year in the past diminished the risk of seasonality impacting the results. Sequentially, bias could have been present in this study, but efforts have been made to minimize bias through a respondent versus non-respondent analysis and the use of a validated questionnaire. Based on these limitations, the study results should be interpreted with caution.

Lastly, the study design did not include a matched control group of children without fractures. By using matched controls, differences in children with and without fractures could have been measured. In this study, descriptive statistical analysis was performed. Therefore, assumptions on the differences between children with fractures compared to the pediatric population referenced in national and international literature can be described and interpreted but not measured. It is strongly advised that future researchers perform a prospective study in a case-controlled setting in order to obtain more reliable results and minimize bias.

A strength of this study was its multi-center setup. Since both a level 1 and level 2 hospital were involved, different patient populations participated in this study. The multi-center setup ensured that this study achieved a sufficient sample size. Additionally, the use of validated questionnaires as a foundation for this study’s questionnaire made the results more reliable. Finally, the questionnaire was kept as simple and short as possible in order to be suitable for participants of varying ages and socio-economic statuses. 

## 5. Conclusions

This study found a relatively high prevalence of adherence to the WHO’s Global Recommendations on Physical Activity for Health among children with fractures. Most fractures were obtained during the children’s participation in sports. This study suggests the need for injury prevention among youth between 13 and 18 years of age and children participating in sports. Future research in a matched case-controlled setting is essential to adequately target injury prevention programs. 

## Figures and Tables

**Figure 1 children-09-00325-f001:**
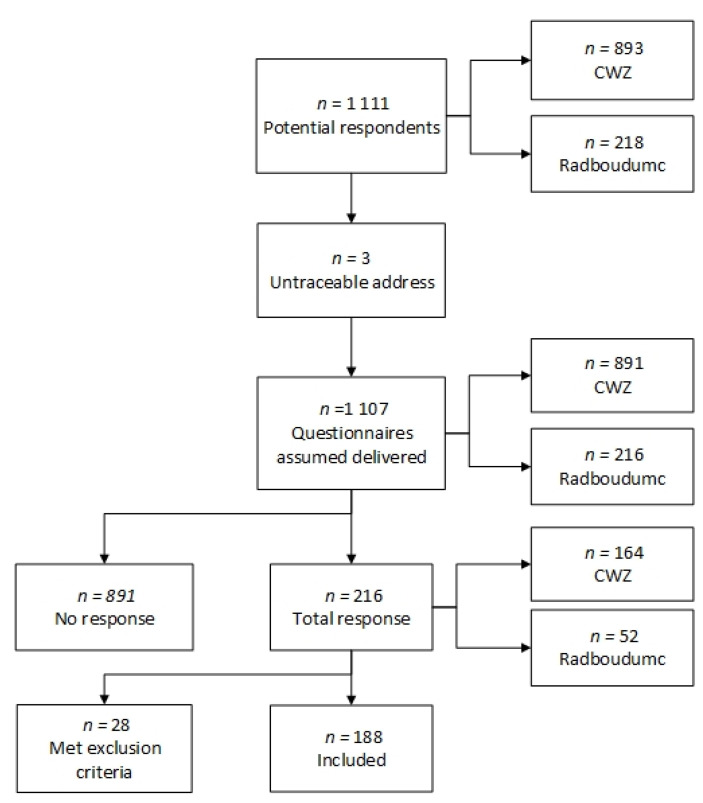
Flowchart of the patient inclusion process.

**Figure 2 children-09-00325-f002:**
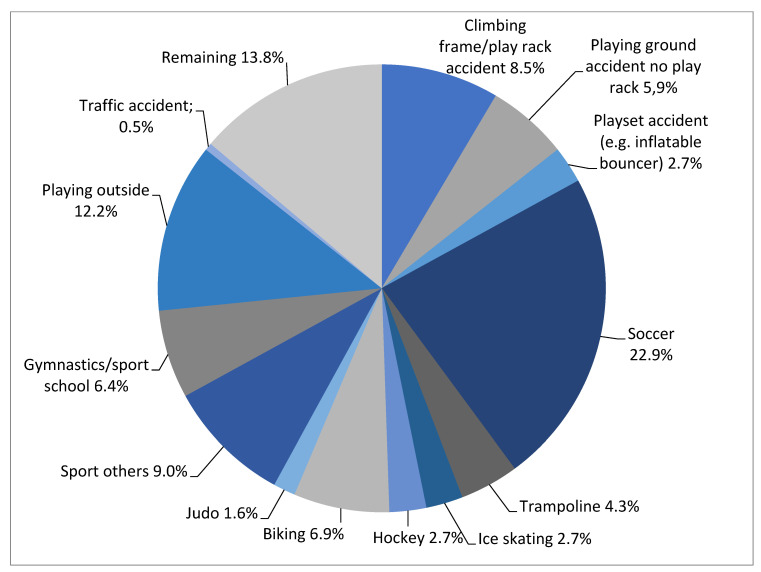
Most frequently occurring mechanisms of trauma (*n* = 188).

**Table 1 children-09-00325-t001:** Baseline characteristics for included respondents vs. non-respondents.

Characteristics	Study Population (*n* = 188)	Non-Respondents (*n* = 891)	*p*-Value
Gender–*n* (%) ^‡^			
Boys	108 (57.4%)	528 (59.3%)	0.646
Girls	80 (42.6%)	363 (40.7%)	
Mean age–years ^†^			
Total	11.2 (SD = 3.7, Range 4–18)	11.2 (SD = 3.9, Range 4–18)	0.816
Boys	11.7 (SD = 3.9, Range 4–18)	11.8(SD = 3.8, Range 4–18)	
Girls	10.6 (SD = 3.37, Range 4–18)	10.2 (SD = 3.7, Range 4–18)	
Age group–*n* (%) ^†^			
4–12 years	109 (58.0%)	520 (58.4%)	0.923
13–18 years	79 (42.0%)	371 (41.6%)	

Used Statistical Test to Calculate Differences between Groups: ^†^
*t*-test; ^‡^ Chi-Square Test.

**Table 2 children-09-00325-t002:** Comparing sufficiently physically active children vs. not-sufficiently active children.

*n* = 188 Children with Fractures	Children Meeting the Recommendations for Health (*n* = 95, 50.5%)	Children not Meeting the Recommendations for Health (*n* = 93, 49.5%)	*p*-Value
Age ^†^ (mean in years)	10.97 (SD = 3.6)	11.49 (SD = 3.8)	0.080
Gender ^‡^ (boy vs. girl *n*, %)	57 vs. 38 60% vs. 40%	51 vs. 42 55% vs. 45%	0.477
BMI *^,†^ (mean)	18 (SD = 3.4)	17.8 (SD = 2.8)	0.332
Screentime ^†^ (mean min/day)	191 (SD = 131.8)	220 (SD = 171.3)	0.384
Type of fracture ^‡^ (Upper vs. lower extremity *n*, %)	79 vs. 16 83% vs. 17%	75 vs. 18 81% vs. 19%	0.654
Trauma occurred during sports **^,‡^ (*n*, %)	56 (53%)	49 (47%)	0.337
Time spent on sports ^†^ (mean in hours/week)	7 (SD = 6.7)	4.5 (SD = 3.2)	**0.004**
Time playing outside ^†^ (Mean min/day)	26.1 (SD = 55.5)	17.8 (SD = 35.2)	0.226

* *n* = 167, missing data *n* = 21 not presented ** *n* = 183, missing data *n* = 5 not presented; *p* < 0.050 are considered significant and presented in bold font. Used statistical test to calculate differences between groups: ^†^
*t*-test; ^‡^ chi-square test.

**Table 3 children-09-00325-t003:** An overview of the study population’s physical activity and trauma mechanism causing fractures.

	Total Population	Boys	Girls	*p*-Value *	4–12 Years	13–18 Years	*p*-Value *
	(*n* = 188)						
**Trauma mechanism** **							
Sports-related	105 (57.4%)	59 (55.1%)	46 (60.5%)	0.468	47 (44.8%)	58 (74.4%)	**0.001**
Fall from 0.5–3 m	52 (28.4%)	37 (34.6%)	15 (19.7%)	**0.028**	40 (38.1%)	12 (15.3%)	**0.001**
Fall from 0.5 m or less	26 (14.2%)	11 (10.3%)	15 (19.7%)	0.071	18 (17.1%)	8 (10.3%)	0.187
**Physical activity**							
Achieved physical activity norm	95 (50.5%)	57 (52.8%)	38 (47.5%)	0.474	61 (56%)	34 (43%)	0.080
Mean hours/day performing intensive activity	2.4 (SD = 1.5)	2.5 (SD = 1.5)	2.4 (SD = 1.5)	0.978	2.6 (SD = 1.4)	2.2 (SD = 1.6)	0.076
Mean hours/week spent on sports	5.8 (SD = 2.5)	6.3 (SD = 6.0)	5.2 (SD = 4.5)	0.165	4.5(SD = 2.8)	7.8 (SD = 7.2)	**0.001**
Mean minutes/day playing outside	37.6 (SD = 56.2)	21.9 (SD = 51.5)	22.1 (SD = 39.6)	0.978	29.7 (SD = 56.3)	11.4 (SD = 2.9)	**0.003**

* *p*-values < 0.050 are considered significant and presented in bold font; ** *n* = 183, missing data *n* = 5 not presented.

## Data Availability

The data presented in this study are available on request from the corresponding author. The data are not publicly available due to privacy considerations.

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
