# Peer review of "Levels of Physical Activity in Children with Extremity Fractures a Dutch Observational Cross-Sectional Study"

_children, 2022, doi:10.3390/children9030325_

Round 1

Reviewer 2 Report

The manuscript called " Levels of physical activity in children with extremity fractures. A Dutch observational cross-sectional study " is an interesting research piece, covering an important aspect related to the associations between the frequency of physical activity and Fractures in children.

 The abstract is attractive and well written, however there is a need to add the number of patients/responders of the study.

The keywords are repeating terminologies already used in the title. In order to avoid repetitions is required to choose different words from the title and the abstract body, amplifying the action of the search engine robots. It is recommend to find other terminologies that can be associated to the paper.  

Introduction:

Introductory section written very well, no unnecessary information. The authors focused on a general description of physical activity (PA) among children and adolescents based on WHO recommendations in comparison with what has been studied about child adolescents so far in the Netherlands. They also briefly but succinctly described the fracture situation among the age category selected for the study.

If using whole expression like physical activity (l.34), please put the abbreviation in brackets (PA) and from that moment on use only the abbreviation. This will apply to any other expression that is used repeatedly in the text.

It would have been worthwhile for the authors to apply research questions that would have reinforced the research undertaken and its explanation in the discussion and conclusion sections.

Methodology and Material

2.1. Study design and setting

Why were these two named facilities chosen for the study? Are these the only ones in the area, or what was the selection key? Additionally, were respondents informed that the survey was voluntary and that they could choose not to respond? What was the response return percentage?

2.2. Study population

The authors mention the gender of the respondents but don’t state in this section what percentage/number of girls and what percentage/number of boys participated in the study.

2.3. Data collection

It is well and clear prepared.

In the figure 1 there is  number 155 below the last left frame. What is it for?

How BMI was calculated? How were the BMI results interpreted? Have relationships such as age and gender been taken into account in the calculations? The composition of body weight, height and age of girls and boys is subject to constant change, so scientists have developed special indicators for children and adolescents by gender - the so-called per(centiles), they are applicable to people aged 2-20 years.

Authors claim: “In contrast, a fall between 0.5 and 3 meters occurred more frequently in children 191

between 4 and 12 years of age compared to youth between 13 and 18 (p < 0.001) (Table 2)”, but I cannot see this in the table they mentioned.

“climbing a frame”? climbing design? (l.195)

Table 3.

In Table 3, under total population there is an error. When we calculate all, given numbers (105+52+26) we come out with 183 and not 188 as the authors state.

Please standardize the notation of p-value to three decimal numbers everywhere.

The results are not sufficiently described. After their analysis, it was not reported/interpreted what the physical activity level of the study children was. Do the authors think it is sufficient or too low or high? There is a lack of explicit response to the results.

Discussion

The discussion is quite well written, although I see information in parts of it, that should rather be in the results section.

One more thing got me thinking after reading the discussion: who filled out the survey for children under school age? Parents/carers? If so, shouldn't this fact be included in the Methodology section and given as a limitation in the Discussion section?

L.243-246; l.259-260; l. 272-274; need references’ support.

References:

Of the 40 references used in the text, half (n=21) are from between 2016-2020. The rest are older items.

It is worth taking a look and adding to the text the latest WHO (2021) records recommending PA.
